# Enhanced Network Intrusion Detection System

**DOI:** 10.3390/s21237835

**Published:** 2021-11-25

**Authors:** Ketan Kotecha, Raghav Verma, Prahalad V. Rao, Priyanshu Prasad, Vipul Kumar Mishra, Tapas Badal, Divyansh Jain, Deepak Garg, Shakti Sharma

**Affiliations:** 1Symbiosis Centre for Applied Artificial Intelligence, Symbiosis International (Deemed University), Pune 412115, India; 2Department of CSE, Bennett University, Greater Noida 201310, India; rv3481@bennett.edu.in (R.V.); pr5987@bennett.edu.in (P.V.R.); pp6781@bennett.edu.in (P.P.); vipul.mishra@bennett.edu.in (V.K.M.); tapas.badal@bennett.edu.in (T.B.); dj9284@bennett.edu.in (D.J.); shakti.sharma1@bennett.edu.in (S.S.)

**Keywords:** anomaly detection, deep learning, intrusion detection system, network security, UNSW-NB15

## Abstract

A reasonably good network intrusion detection system generally requires a high detection rate and a low false alarm rate in order to predict anomalies more accurately. Older datasets cannot capture the schema of a set of modern attacks; therefore, modelling based on these datasets lacked sufficient generalizability. This paper operates on the UNSW-NB15 Dataset, which is currently one of the best representatives of modern attacks and suggests various models. We discuss various models and conclude our discussion with the model that performs the best using various kinds of evaluation metrics. Alongside modelling, a comprehensive data analysis on the features of the dataset itself using our understanding of correlation, variance, and similar factors for a wider picture is done for better modelling. Furthermore, hypothetical ponderings are discussed for potential network intrusion detection systems, including suggestions on prospective modelling and dataset generation as well.

## 1. Introduction

Cyberspace currently has become increasingly complex with the sheer number of assets deployed and devices being used. It is very hard to delineate the exact variables accurately to be able to predict if an attack has occurred within an environment of systems. Network intrusion detection systems (NIDS) have historically been used as a form of network forensics reporting anomalous events and alerting the relevant authorities. Commonly known NIDSs are based on signatures in terms of patterns of events and/or executable files or anomalies in events [1]. Very recently, in May 2020, we saw 8,801,171,594 breaches in data records [2]. These events suggest that even the smallest of breaches can be very damaging, and thus, the need for a sensitive detection system arises in such a dire situation. We might see an even worse increase in the upcoming months due to the rapidly increasing connectivity of assets and devices in general.

To detect such events, corporations and households require network forensics to understand the specific flows causing breaches and thereby curb any continuation of attacks. Many older datasets relevant to the construction of NIDS have proven to be insufficient for real-time scenarios. Therefore, the cybersecurity research group at the Australian Centre for Cyber Security (ACCS) and other researchers of this domain around the globe had taken up the challenge to produce a good enough dataset that would work for real-time environments effectively. Thus, a dataset called the UNSW-NB15 Dataset was constructed. Therefore, we aim to create a model for a NIDS that predicts anomalous events, i.e., attacks accurately with low misclassification error using this dataset.

The rest of the paper shall be organized as follows: In Section 2, we shall be looking at the past work done on modelling a NIDS using the UNSW-NB15 Dataset and the various techniques they have employed and their results. In Section 3, we will look more into the details surrounding the UNSW-NB15 Dataset in terms of its construction, features, labels, and evaluated metrics. In a separate section, the preprocessing is done before modelling. In Section 4, we shall investigate the models being used and their specific technical definitions with respect to the understanding of the dataset as laid out in Section 3. Section 5 will explore the results we have found and the various kinds of metrics involved for cross-validation. In Section 6, we shall propose other possible methodologies for modelling and hypothetically creating more datasets like UNSW-NB15 as a serving guide for further research on NIDS/NIPS construction. Finally, we conclude our discussions with the results we found in Section 7.

## 2. Literature Review

In [3], the authors implemented a scheme in which they applied network forensic analysis for sniffing packets at edge devices and stored said packets in large volumes in the SQL database. They used the UNSW-NB15 Dataset for packet-feature analysis; however, their model is reasonably weaker. The selected features used in the model for network forensic analysis are inadequate for accurately predicting attacks on the system. Hence, unlike our model, it may leave loopholes of breaches in the NIDS system due to information loss. Additionally, accessing large scale data from the SQL database is extremely slow compared to our locally recorded packets. Nour Moustafa et al. Created a network threat intelligence system for something called Industry 4.0. Their technique primarily uses Bro-IDS for packet capture and focuses mainly on product-based IoT devices and machinery used by the industries and consumers. As the name suggests, “Industry 4.0” represents the emerging industries and start-ups in the IT field aiming to bring about revolutionary changes to technology. Accordingly, they planned to move their system to the cloud. They attempted to construct an architecture of interconnectivity through multiple edge devices through internet bridges. These devices are naturally IoT devices. They use MHMM (mixed hidden Markov models) and other complex models for threat detection with their system’s carefully laid out architecture. However, their system’s major drawback is a lot of public-end network devices and an inadequacy in specifying the device-respective firewalls, which may lead to complex MITM (man in the middle) attacks by experienced hackers/crackers. Such complex models take a lot of time to get debugged as a huge amount of middleware is involved in the cloud system as well. Any breach of data or damage in hardware will not be easily resolved on time compared to our end-to-end architecture which is fairly simpler and has a lot more scalability, with the added benefit of less exposed public devices in the framework [4].

In [5], the authors created a multilayer feedforward artificial neural network. The model was trained on the UNSW-NB15 Dataset with a few modifications. The model was evaluated using various experiments on the modelling architecture, and the results were impressive. However, their finalized model is computationally expensive, so it takes a lot of time and resources to train the model and later run it, which is not ideal for detecting attacks as soon as possible. In [6], the authors proposed a BMM (beta mixture model) based ADS (anomaly detection system), which is a probabilistic representative model of a subset of the entire dataset. The probability distribution in this case can adapt to arbitrary variables with different ranges. They conducted experiments on the UNSW-NB15 Dataset to evaluate the performance of their model. While being an adaptive model, the distribution relies on some uniformity in the subset. It, therefore, is not able to reach greater heights in terms of detection rates and false-positive rates.

In [7], Preeti et al. implemented the concept of “cloud hypervisor” for the security-based architecture using a software intrusion detection system involving different end devices consisting of software connected to the internet and the cloud. They have considered the security in the cloud to be of great significance. Hence, they have used the PSI-NetVisor hypervisor layer to deploy a virtual machine introspection (VMI) functionality to perform network monitoring and the related process execution tracing involved with it. It also detects attacks using a behaviour-based intrusion detection system (BIDM). However, the model is robust towards the connected end-devices to the cloud. Even though the cloud-based security is maintained and the same dataset is used as ours, many devices running software are connected to a third-party CSP and can be breached if malware is injected into any of the VMs, unlike our case where the model is run independently of the hypervisor layer. They did not place a good amount of significance in modelling for the prediction of attacks but rather focused their efforts in the architecture of their system and thereby failed to make their entire system widely implementable. In [8], The authors focused on the problem of huge network volumes, imbalanced datasets, and other outdated datasets for different NIDSs, leading to a weaker system for predicting attacks and loopholes in network security. Hence, they constructed a robust model to data imbalances and utilized a sparse multiclass classification approach known as Ramp-KSVCR (ramp K-support vector classification regression). This support vector classifier was mainly used to remove noise and other outliers in the dataset for better generalizability. They have also used the modern UNSW-NB15 Dataset for prediction. Our model secures an upper hand, however, since instead of eliminating outliers, we work with them and avoid a loss of information due to our intuition that these very outliers may be highly indicative of event anomalies instead.

In [9], the authors proposed an aggregator model based on the theory of flow-level analysis. Their model uses simple random sampling and ARM (association rule mining) to select observations for IDS (intrusion detection system). The ARM method relies on the association of features regarding the frequency of a preceding event in case an antecedent event has occurred. They focus mainly on modelling based on correlative measures as they believe that is the more effective way of modelling an IDS. However, they focus too much on the relevancy of flows than on the predictive ability. They use a very small subset of features to decrease the processing time and in turn, lose predictive performance and generalizability. While our model certainly would lag behind such a model in terms of performative speed, it compensates for predictive power. In [10], the authors have addressed the problems of data breaches and interceptions within an interconnected set of devices in a public network. To mitigate such risks, they have proposed the idea of novel GAA (geometric area analysis). This technique focuses on calculating the respective areas of the features calculated from the beta mixture model parameters, and they have assumed that the network samples are *iid* (independent and identically distributed). To reduce the false-positive rate of their model, they have applied PCA over the UNSW-NB15 Dataset. However, this preprocessing step leads to a problem while modelling—i.e., having lesser features for calculating the area, leading to information loss. Meanwhile, our model does not eliminate any feature except for creating a feature out of a couple of features and achieves a reasonable false-positive rate without resorting to decreasing dimensionality.

In [11], the authors have mainly focused on detecting zero-day attacks on the systems. They have tried to work on this by focusing on reducing the false-positive rate using ADS (anomaly detection system). They have used the NSL-KDD and UNSW-NB15 Dataset for training their model and use finite Dirichlet mixture models, maintaining a normal dataset so that there is no occurrence of under fitting in the model. Achieving this on a large scale is quite a challenge as cloud systems have nodes that cannot be identified as centralized or distributed. Therefore, high-speed transfer of data among nodes may affect the performance of ADS in terms of false-positive rates and detection rates.

In [12], the authors have addressed botnet activities in their study on IoT (internet of things) by using machine-learning algorithms such as decision tree, association rule mining model, artificial neural networks, and naive Bayes classifiers. They got their best results using a decision tree for the binary classification problem, and they also ran their models on the NSL-KDD Dataset and found a better model as compared to using the UNSW-NB15 Dataset. They tried to pair these models with network forensics in IoT. They primarily worked on these models to eliminate botnets in IoT, which is currently a huge problem in this age of digital interconnectivity. However, since NSL-KDD is not a sufficiently good dataset for modern-day attacks, their poorer performance on the UNSW-NB15 Dataset is problematic.

In [13], authors have utilized machine learning and deep learning approach to solve intrusion detection using heterogeneous dataset. Spark MLlib based robust classical ML classifiers for anomaly detection and convolutional-auto encoder for misuse attack. Moreover in [14], the authors worked on quick detection of intrusion for that authors proposed swiftIDS which utilized light gradient boosting machine to handle the massive traffic data. In another study [15], the authors targeted detection of intrusion using decision tree algorithm, and rule based concepts for securing internet of thing by classifying network traffic as attack or benign.

## 3. Descriptive Analysis of the Dataset

### 3.1. Dataset Description

The ACCS uses the IXIA PerfectStorm tool to simulate normal and abnormal events. The tool contains all the relevant information about modern attacks since their production of events is dependent on the updates on the CVE site. This site is a very well-known repository for discovered exploits, vulnerabilities, and breaches. To generate features for the flow of packets, the tools TCPDump, Argus, and Bro-IDS were used primarily alongside a couple of C# algorithms to generate relevant information about nuances within the flows of packets [16]. Initially, the features given to us are listed in Table 1, Table 2, Table 3 and Table 4 along with their descriptions [17].

The features in Table 1 correspond to categorical variables that determine details on the source to destination traversal of packets in terms of IP Addresses, protocols used and inadvertently the state used within the protocol and the service used. However, features like srcip and dstip are not used for modelling since they are more suited for prevention measures rather than for detection.

The features in Table 2 correspond to discrete integer values representing information on sub-details of protocol implementation and the number of connections/flows associated with similar source/destination attributes. Features like these are very important while determining an attack because if a cluster of packets is being sent to a singular place, it highly indicates suspicious activity. Coupling features like sttl and dttl, in combination, create a better schema for detecting attacks in a model.

The features in Table 3 represent continuous values representing temporal details surrounding packet flows/connections, such as rates as bits per second and the list of durations of information exchange between request–response transactions. Features like Sload and Dload are often helpful in determining unusual inbound data rates and thereby arousing suspicion. The round-trip times for TCP connection setups are also very relevant since it provides information on an attacker’s attempts to reverse connect into the system. Other features may affect, but it is expected for them to be underwhelming. It is important to note that features like these might have a linear relationship with the labels and thus need to be standardized to one single scale to prevent instability in prediction.

The features in Table 4 are either timestamp of the starting time and ending time of one event (packet flow/connection) or binary values representing true/false values for whether an event happened or not. An interesting indication for an attack in the event of an FTP login occurring using username and passwords. Normally, FTP servers are disabled in modern times unless necessary due to it being a security hole. However, in the dataset, this feature was very sparse in nature. Still, it was certainly expected of it to be somewhat useful for prediction in terms of arousing an intuitive suspicion of an attack.

### 3.2. Dataset Preprocessing

The dataset is comprehensively explained in [16] and was used as a basis for understanding the dataset. The dataset was given in the form of four raw CSV files, namely, UNSW-NB15_1.csv, UNSW-NB15_2.csv, UNSW-NB15_3.csv, and UNSW-NB15_4.csv. We combined the raw four CSV files and did the preprocessing and modelling on the final dataset. The preprocessing went as follows:(1)Firstly, we removed the duplicate entries and an entry containing an event with a localhost IP Address.(2)Then we dropped the features srcip, dstip.(3)Replaced the ‘-’ and whitespace entries with a NaN(4)In the columns sport and dsport, there were hexadecimal values instead of integer values, so we converted them accordingly. For example, 0x000c was converted into the integer equivalent, that is, 12.(5)We made a new feature called duration, which is the difference between Ltime and Stime and therefore removed Ltime and Stime.(6)The columns proto, state, and service were label encoded.(7)A separate dataset was constructed where proto, state, and service were one-hot encoded instead. This dataset was constructed for neural network modelling.(8)The entire dataset’s missing values were imputed using different k-nearest neighbour imputers of varying parameters (in terms of the number of neighbours, which are proportional to the number of missing values while imputing).(9)attack_cat was also dropped since we are only modelling if the event is an attack or not.(10)Z-score normalization of features that had relatively continuous values was done.

### 3.3. Dataset Analysis

Looking at the correlation matrix of the resultant Dataset after preprocessing as shown in Figure 1, the correlations of features with the labels were relatively high for sttl, dttl and ct_state_ttl and intermediately high for Sload, tcprtt, synack, ackdat, ct_src_dport_ltm, ct_dst_sport_ltm, and ct_dst_src_ltm.

There were many low variance features, but removing them would result in a loss of information during modelling, so we decided against removing them since it would be more efficient to model for sparsity instead since sparsity is very common, generalizability in a realistic context is better achieved. Additionally, many features held reasonable importance when we checked for their correlation with the labels as well. Our new dataset has 1,959,770 labels for no attack occurring and 99,643 labels for an attack occurring. Therefore, the labels that we are interested in comprise 4.83% of all the entries in the dataset. This is a very imbalanced dataset, and therefore, we needed to create our model accordingly. While checking for outliers, we saw a very high correlation of outliers within features with the anomalous output label ‘1’, so we decided against removing outliers as they would be very crucial in our anomaly-predictive modelling

In summary, we had a problem of relative sparsity in feature values and an imbalance in the number of labels. To combat these problems, we try various kinds of models and see which works the best.

## 4. Classification Modeling

In this study, we implement eight kinds of models—i.e., neural network, naive Bayes classifier, SVM, decision tree, extreme gradient boosted tree, random forest classifier, AdaBoost classifier, and logistic regression models. All of these models were implemented in CPython compilers for temporal and memory efficiency and were run on an NVIDIA GeForce GTX 1050 Ti 4GB GPU and Intel i7 7700HQ for models that were not possible with GPU acceleration. Random states were used for each model to avoid overfitting when run through cross-validation.

### 4.1. Evaluation Metrics

The metrics used to evaluate model performance are accuracy, DR (detection rate), FAR (false alarm rate), ROC-AUC score, precision, recall, and *F*_1_ score for both the training and testing splits and their definitions are as follows.

*Accuracy* is the rate of the truly classified data to all the data [18,19,20,21,22], which is denoted as.
(1)$$Accuracy=\frac{TP+TN}{TP+TN+FP+FN}\;$$
where *TP* represents the number of true positives and *FN* represents the number of false negatives *TN* represents the true negative and *FP* represents false positive.

False alarm rate (*FAR*) is the percentage of incorrect predictions made of anomalous events out of all predictions made.
(2)$$FAR=\frac{100\times FP}{\left(FP+TN\right)}\;$$
where *FP* represents the number of false positives and *TN* represents the number of true negatives.

ROC-AUC score is the area under the receiver operating characteristic curve that calculates true and false positives at different classification thresholds [20].

*Precision* is the percentage of relevant predictions made out of all predictions. ‘Relevant’ here means the positively labelled entries.
(3)$$Precision=\frac{TP}{\left(TP+FP\right)}\;$$

The *Recall* is the sensitivity of the model of outputting relevant predictions
(4)$$Recall=\frac{TP}{\left(TP+TN\right)}\;$$

*F*_1_ score is the collative score generated via the results from precision and recall and is understood as a single-valued metric for the performance over specific class labels [20].
(5)$$F_{1}=2\times \frac{Precision\times Recall}{Precision+Recall}\;$$

### 4.2. Model Definitions

#### 4.2.1. Artificial Neural Network

The neural network has five hidden layers with the output as a softmax unit for two classes. The unit is mapped for each neuron mathematically defined in Equation (6).
(6)$$\mathrm{softmax}\left(x_{i}\right)=\frac{e^{\left(x_{i}\right)}}{\sum_{j}e^{\left(x_{i}\right)}}$$
where *x* is a single-valued output from a list of multi-classed predictions

The model was trained on the dataset with one-hot encoded columns instead since with neural networks; we can simultaneously reduce dimensions in later layers and produce a classifier. The kernel is initialized with Lecun’s initializing method using a truncated normal distribution centred on 0 as illustrated in [9] since it is required to be used by the SeLU activation function used in model. In this network, 140 Neurons are used for each hidden layer with dropout layers after every hidden layer with the ranges of (0.06–0.22). The dropout layers act as a preventive measure for overfitting as well as emulating the intuition of dropping irrelevant information arising from the increased dimensionality of the data for improved classification. Each hidden layer uses SeLU activation [23] as mathematically defined in Equation (7).
(7)$$f\left(a,x\right)=\lambda \left\{\begin{array}{r} \alpha \left(e^{x}-1\right),\;x\geq 0\\ x,\;x<0 \end{array}\right.$$
where *x* is an input value from the layer before the current layer and *α* = 1.67~ and *λ* = 1.05

The model was compiled with a categorical cross-entropy loss [24] as given in Equation (8) since our last layer is a softmax unit and fitted using the Adam optimizer on default hyperparameters. The batch size was set to be 2048 and the learning rate as 0.0012. These hyperparameters were decided after experimentation with an early stopping mechanism that monitored loss over 12 epochs with the best weights restored. Progress of this model was tracked on Tensorboard using Tensorflow callbacks.
(8)$$cross\;entropy=-\overset{e=1}{\underset{M}{\sum }}y_{o,c}\log\left(p_{o,c}\right)$$
where *M* is number of classes, *y_o_*_,*c*_ is class label c for observation o, $p_{o,c}$ predicted probability of class *c* for observation *o*.

#### 4.2.2. Naive Bayes Classifier

A gaussian-based naive Bayes classifier was used with variance smoothing as 10^−9^ and assumed the likelihood of features mathematically defined in Equation (9)
(9)$$P\left(x_{i}|y\right)=\frac{1}{\sqrt{2\pi \sigma_{y}^{2}}}\exp(-\frac{{\left(x_{i}-\mu_{y}\right)}^{2}}{2\sigma_{y}^{2}})$$
where *x* is your current input, *y* is the previously calculated vactor in the chain of probabilities, *σ* is the standard of *y*, *μ* is the mean of *y.*

#### 4.2.3. SVM

A stochastic gradient descent classifier was used in conjunction with a linear SVM with a squared hinge loss as defined in Equation (10).
(10)$$L\left(y,\hat{y}\right)=\sum_{i=0}^{N}\left(\max{\left(0,1-y_{i}.{\hat{y}}_{i}\right)}^{2}\right)$$
where *y* is the vector containing the actual values and *ŷ* is the vector containing the predicted values.

A squared hinge loss was used to penalize wrong predictions more than a typical hinge loss for better performance in terms of anomaly prediction. However, to generate a ROC-AUC Score, the model was passed through a calibrated classifier to generate predicted probabilities.

#### 4.2.4. Decision Tree

A decision tree with a maximum depth of nine levels was used. The Gini method was used for splitting nodes, with the minimum number of samples split being 2. We also set the class weights inversely proportional to the class frequencies in the input data to provide a stricter condition to misclassifying the least frequent classes.

#### 4.2.5. Extreme Gradient Boosted Tree

In [25], an approximate algorithm is mentioned for finding splits within nodes and was thus used for modelling since the dataset used is spatially extensive. An extreme gradient boosted tree is used with 350 estimators with the scale positioning weight as the ratio of the number of negative instances to the number of positive instances to accordingly adjust the weightage of predictions according to the number of classes present within the input. The maximum depth was set to 9, to avoid overfitting to the input with a softmax loss function as shown above in Equation (6). The learning rate was set as 0.3 with subsampling so that we prevent overfitting to a particular set of training instances. These hyperparameters were set in a model created via XGBoost, a library for gradient boosting libraries, and the model was trained using GPU.

#### 4.2.6. Random Forest Classifier

A random forest classifier with no maximum depth (all leaves expanded with no pruning) was implemented with 300 estimators with other hyperparameters similar to the earlier mentioned decision tree classifier.

#### 4.2.7. AdaBoost Classifier

An AdaBoost classifier was used with its base estimator as a random forest of three estimators. This entire classifier had a total of 200 estimators. An algorithm called “SAMME. R” was used, which is a regularized version of the algorithm mentioned in [26] and is used since our base estimator produces continuous values in terms of label probabilities. It also converges faster than other algorithms.

#### 4.2.8. Logistic Regression

A logistic regression model was fitted using the non-linear conjugate gradient method as illustrated in [27] over maximum iterations of 300 with an L2 penalty on weights to prevent overfitting. We experimented with other kinds of penalties, but the L2 norm helps us penalize misclassification of anomalies.

### 4.3. Methodologies

The models were trained over five stratified folds for cross-validation. The splits were made randomly using different seeds, and the feature importance’s are calculated using the F1 scores with respect to individual weights for the best performing model. Early stopping was implemented with respect to the observed change in losses for 10 values, and the best sets of weights are restored for the artificial neural network. All eight models were experimentally evaluated based on the bias-variance trade-off concerning training and testing accuracies and the precision, recall, and F1 scores for each class within our label set. Uncertainty analysis using the stratified folds on our training and testing with different seeds was done, and the mean and the standard deviation of the resultant training/testing accuracies were noted down for evaluation. Consequently, the detection rate and false alarm rate for attacks were also noted. The ideal condition is of a very high detection rate contrasting with a very low false alarm rate.

## 5. Results and Discussion

After evaluating all the necessary metrics for all the models, our final model for the NIDS turned out to be the extreme gradient boosted tree which was generated using XGBoost.

In Figure 2, the top 15 features are plotted, and interestingly, sbytes had the highest F Score. As expected, dsport, sport, and proto also have a good amount of importance.

In Table 5, we were able to generate a low bias–low variance result for all models. The standard deviation for all models is also very low for different train-test splits despite the relative sparsity of data for different shuffled splits.

In Figure 3, comparing Adaboost and XGBoost, we can see that XGBoost has a higher detection rate and lower false alarm rate than all other models for both training and testing sets and is a clear winner with respect to these metrics. There were very small differences between the calculated false alarm rates. Still, we could see a noticeable difference between the training and testing detection rates, and it was interesting to see that the detection rates were higher for unseen data. This would suggest good generalizability when given a lesser amount of unseen data.

Since we are mostly interested in whether the model performed well in terms of sensitivity towards events with attacks. We estimated the precision, recall and final F1 score for the ‘Attack’ label in Figure 4. These scores represent a subset of predictions where the false positives represent the event where anomalous events are predicted as normal events. This comes under the category of missing alarms and is especially dangerous if these numbers are not up to the mark. Higher the overall F1 score, better is the model’s capability to avoid missing alarms. Following these results, we can observe that XGBoost does a lot better on an unseen dataset as compared to other models, which seem to have ‘overfitted’ in terms of sensitivity.

In Figure 5, we can see that all models perform very similarly in terms of classifying events as normal events due to the larger amount of data on normal events. This is why an ideal model is important to capture attacks more often and report falsities less often. Therefore, the results obtained in Figure 3 become more crucial in our evaluation for each model. In summary, most models performed well in terms of detecting attacks and fewer misclassification errors, so the minute differences were what made XGBoost distinctive. The performance of XGBoost barely had any difference while running the model on testing sets compared to the training sets.

In Figure 6, we can observe that XGBoost achieves a higher ROC-AUC score than its counterparts for both splits. This extremely high result indicates that irrespective of a classification threshold chosen, our model can classify positive results with a very high percentage, in this case, 99.99 for the training set and 99.95 for the testing set. While AdaBoost predictive power is higher in comparison for the training set, it compromises that power with not being able to generalize to unseen data as well as its counterparts.

Our proposed model outperforms all other models referenced from the literature by a huge margin, as shown in Table 5 and Table 6. Moreover, detailed result obtained from proposed XGBoost model have been shown in Figure 7.

## 6. Further Research

Since the dataset collectively is data-intensive, one can also use auto-encoder models for anomaly prediction for better generalizability. The dataset, while being data-intensive, lacks real-time data even though it simulates modern attacks very well, considering the way it is constructed. Despite the construction involving compiling 100 GBs of packet flows, it still lacks information on real-time attacks where hackers do not utilize typical attacks but instead utilize zero-day exploits. The human element of malicious strategies is also not accounted for in this dataset. Therefore, we believe that an ensemble of models should focus on packet flows, changes in asset inventory, data downloaded, and/or uploaded, and other similar operations that may occur in corporate organizations for a better NIDS. This hypothetical ensemble would cover up for the lack of information on human agency to a good extent. For example, we can ensemble the model proposed in this paper with the other models which would work on other functionalities such as detecting anomalous changes in asset inventory merge them with a SIEM which would act as a form of artificial intelligence, with respect to cyber defence and thus evolving a series of NIDs into one big NIPS (network intrusion prevention system). This idea, however, is ambitious and needs work not only on the individual models but also on the meshing of these models along with deployment mechanisms.

## 7. Conclusions

In conclusion, this paper demonstrated that—on the evaluation of various models—the extreme gradient boosted tree stood out of them all and has performed reasonably well as compared to other models in the literature. Our model gave us an excellent performance as expected from an ideal NIDS. Our model achieves an accuracy of 99.64%, a detection rate of 95.47%, and a very low false alarm rate of 0.012% on unseen data.

## Figures and Tables

**Figure 1 sensors-21-07835-f001:**
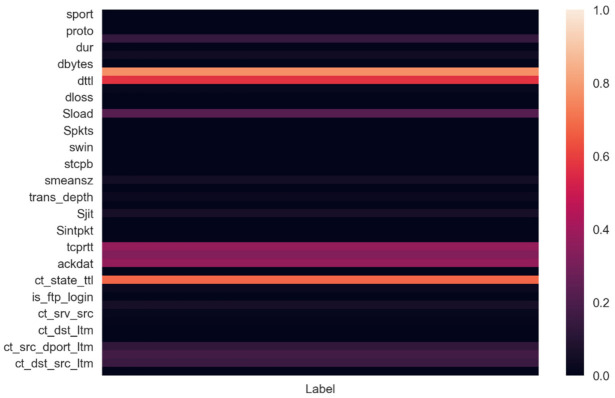
Correlations of the highest ranked features with output labels.

**Figure 2 sensors-21-07835-f002:**
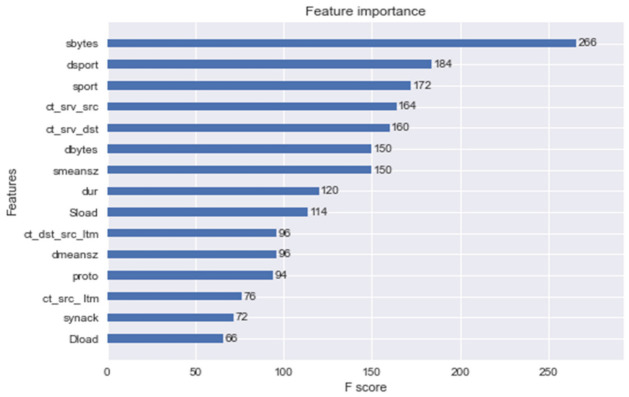
Feature importance generated by XGBoost.

**Figure 3 sensors-21-07835-f003:**
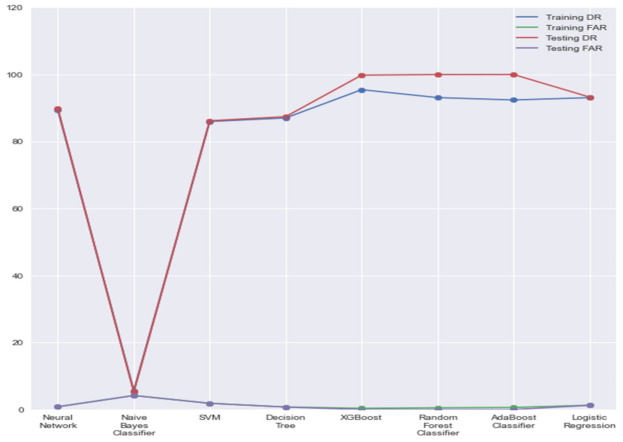
DRs and FARs of all models.

**Figure 4 sensors-21-07835-f004:**
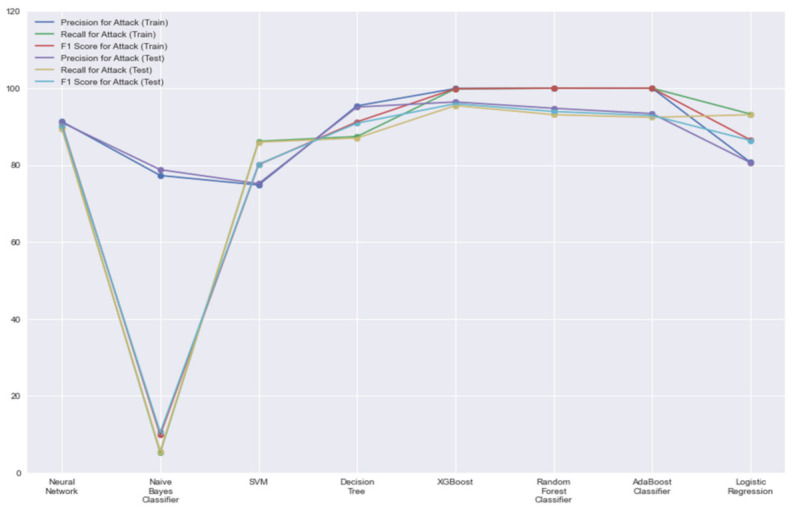
Classification report for ‘Attack’ label.

**Figure 5 sensors-21-07835-f005:**
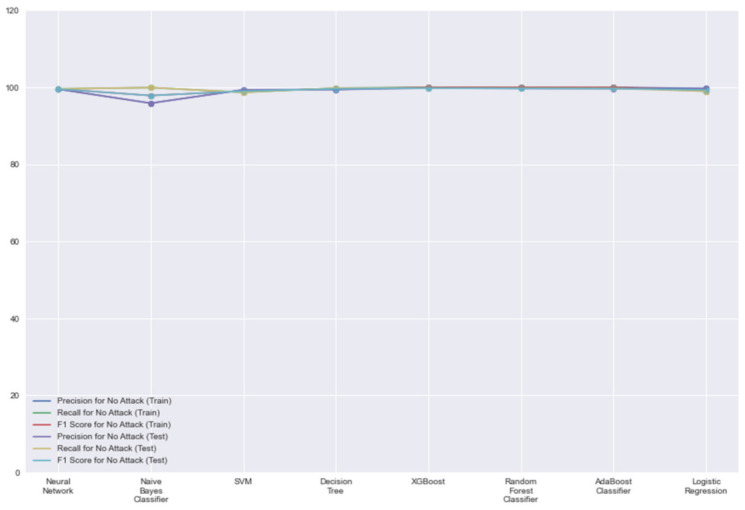
Classification report for ‘No Attack’ label.

**Figure 6 sensors-21-07835-f006:**
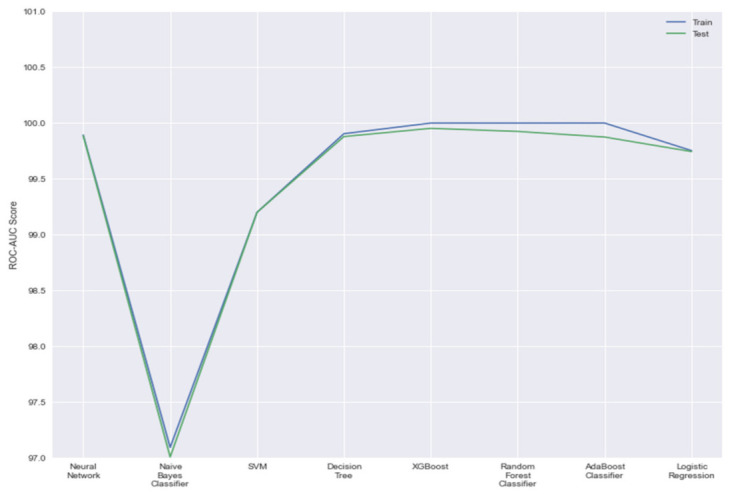
ROC-AUC scores of all models.

**Figure 7 sensors-21-07835-f007:**
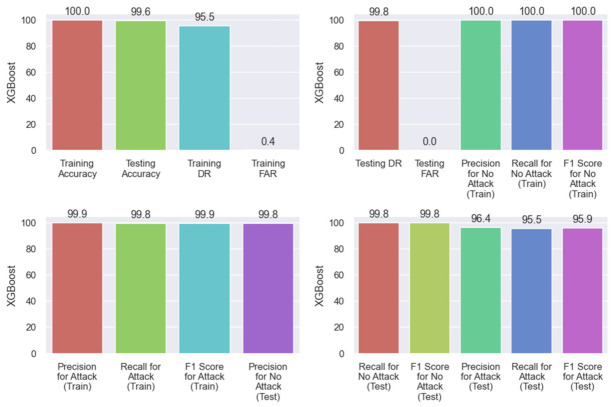
XGBoost scores.

**Table 1 sensors-21-07835-t001:** Nominal features.

Feature Name	Description
srcip	Source IP address
dstip	Destination IP address
proto	Transaction protocol
state	The state depending on the protocol for the event (for individual packets)
service	Service used w.r.t the transaction of packets. Such as HTTP and FTP.

**Table 2 sensors-21-07835-t002:** Integer-based features.

Feature Name	Description
sport	Source port number
dsport	Destination port number
ct_srv_src	No, of times the service and the source address were the same in the last 100 connections
ct_srv_dst	No, of times the service and the destination address were the same in the last 100 connections
ct_dst_ltm	No. of times, the destination address was the same in the last 100 connections.
ct_src_ ltm	No. of times, the source address was the same in the last 100 connections.
sbytes	Amount of bytes from source to destination
dbytes	Destination to source transaction bytes
sttl	Source to destination time to live value
dttl	Destination to source time to live value
sloss	Source packets retransmitted or dropped
dloss	Destination packets retransmitted or dropped

**Table 3 sensors-21-07835-t003:** Float-based features.

Feature Name	Description
dur	The total duration of flow
Sload	Source to destination bits/s
Dload	Destination to source bit/s
Sjit	Jitter from source to destination
Djit	Jitter from destination to source
Sintpkt	Inter-packet arrival time from source to destination
Dintpkt	Inter-packet arrival time from destination to source
tcprtt	Sum of ‘synack’ and ‘ackdat’
synack	The time between the SYN and the SYN_ACK packets.
ackdat	The time between the SYN_ACK and the ACK packets.

**Table 4 sensors-21-07835-t004:** Binary and timestamped features.

Feature Name	Description
is_sm_ips_ports	When the source and destination IP Address and the port are the same, the value is 1 else 0
is_ftp_login	If an FTP session is opened with the proper credentials, then the value is 1 else 0
Stime	Starting timestamp for the event
Ltime	Ending timestamp for the event
Label	0 for no attack and 1 for attack

**Table 5 sensors-21-07835-t005:** Model accuracies.

Models	Training Accuracy	Testing Accuracy
Neural Network	99.189 ± 1.111 × 10^−6^	99.153 ± 1.111 × 10^−6^
Naive Bayes Classifier	95.838 ± 3.154	95.804 ± 2.154
SVM	98.149 ± 2.66 × 10^−1^	98.143 ± 1.55 × 10^−1^
Decision Tree	99.274 ± 1.13 × 10^−17^	99.235 ± 1.13 × 10^−17^
XGBoost	99.987 ± 2.12 × 10^−16^	99.646 ± 2.32 × 10^−17^
Random Forest Classifier	99.999 ± 3.21 × 10^−15^	99.473 ± 3.51 × 10^−15^
AdaBoost Classifier	100.0 ± 7.78 × 10^−7^	99.381 ± 7.97 × 10^−7^
Logistic Regression	98.745 ± 0.567	98.719 ± 6.67 × 10^−1^

**Table 6 sensors-21-07835-t006:** Comparison of our proposed model with the models proposed in the literature.

Algorithm	Accuracy (%)	FAR (%)
Decision Tree Classifier [18]	85.56	15.78
Logistic Regression [18]	83.15	18.48
Naive Bayes Classifier [18]	82.07	18.56
ANN (Single Layered) [18]	81.34	21.13
EM Clustering [18]	78.47	23.79
Ramp-KSVCR [8]	93.52	02.46
PSI-NetVisor [7]	94.54	02.81
CANID [19]	99.36	-
Proposed Model (XGBoost)	99.95	0.4

## Data Availability

Data supporting reported results can be found on https://github.com/raghavverma651/NIDS-Using-UNSW-NB15 (accessed on 11 November 2021).

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
