# Peer review of "Enhanced Network Intrusion Detection System"

_sensors, 2021, doi:10.3390/s21237835_

Round 1
Reviewer 1 Report
The paper discusses an important topic that is related to anomaly detection. The paper reviews the development of a model using NIDS 53 data sets that accurately predict anomalous events with low misclassification error using this dataset. The authors examined the performance of various models for Network Intrusion Detection Systems (NIDS) using evaluation metrics that involve the calculation detection rate and false alarm rate.
Traditionally, anomaly detection is followed by a decision process, always performed with uncertainties related to False-negatives and False positives. From a statistical point of view, if the true positive is related to the presence of an event and the true negative is associated with the absence of an event, then a false positive is an outcome where the model incorrectly predicts the true positive class, which leads to false alarms (FA). In contrast, a false negative is an outcome where the model incorrectly predicts the negative class, which leads to missing alarms (MA). Therefore, MA and FA are two possibilities that go against each other. In other words, if FA is reduced, the MA increase and vice versa.
The authors consider the accuracy of anomaly detection by a high detection rate and low FA rate, which is an incorrect statement and consideration. As stated above, while the FA rate is reduced, the MA rate will increase accordingly; thereby, the detection won’t be accurate. Therefore, the authors should discuss the evaluation metrics based on a trade-off between FA and MA for optimal detection.
Hence, the reviewer is concerned about using the performance metrics in eq. 1 (line 277) and in eq. 2(line 283). The authors should revise the evaluation metrics section by providing more explanations and supporting references. Note that the decision concept has been widely defined in signal detection theory, see for instance [1].
Below are some minor revisions:
- What NIPS in line 22?
- The authors should revise the Figures' numbers to match with the citations in the text.
[1] Signal Detection Theory, Vi͡acheslav Petrovich Tuzlukov, Springer Science & Business Media, 2001 - Mathematics - 725 pages
Author Response
Original Manuscript ID: sensors-1397879
Original Article Title: “Enhanced Network Intrusion Detection System”
To: Editor
Re: Response to reviewers
Dear Editor,
Thank you for allowing a resubmission of our manuscript, with an opportunity to address the reviewers’ comments.
We are uploading (a) our point-by-point response to the comments (below) (response to reviewers), (b) an updated manuscript with yellow highlighting indicating changes, and (c) a clean updated manuscript without highlights (PDF main document).
Best regards,
Ketan Kotecha et al.

Reviewer 2 Report
The authors discuss IDS for attack detection. This topic is topical and intensively studied. A number of both scientific studies and commercial solutions exist these days and artificial intelligence methods are a common part of them. The authors structure the text well and the section that discusses the current state of the art is well written in my opinion. However, the publications cited do not, in my opinion, give a complete picture of the possibilities of building IDS, but otherwise I find this section acceptable. However, what I think is the main problem of this text is the presentation level from the part that is supposed to present the methods used for the analysis in IDS. The level here is very low. Describing metrics that are commonly used to evaluate the performance of classifiers is possible, but probably unnecessary given how briefly each method is then presented. Why did the authors feel the need to provide a formula for softmax and not, say, for cross entropy calculation? The neural network and its structure would deserve a better description. The other classifiers are mentioned in a few lines, including the one (Extreme Gradient Boosted Tree) that was the worst of all in the subsequent comparison of results obtained. Therefore, I highly recommend improving the accuracy of this text before it should be pulbiked.
Author Response

(The authors gave the same response as above.)
